# Based on Tau PET Radiomics Analysis for the Classification of Alzheimer’s Disease and Mild Cognitive Impairment

**DOI:** 10.3390/brainsci13020367

**Published:** 2023-02-20

**Authors:** Fangyang Jiao, Min Wang, Xiaoming Sun, Zizhao Ju, Jiaying Lu, Luyao Wang, Jiehui Jiang, Chuantao Zuo

**Affiliations:** 1Department of Nuclear Medicine and PET Center, National Center for Neurological Diseases and National Clinical Research Center for Aging and Medicine, Huashan Hospital, Fudan University, Shanghai 200235, China; 2Institute of Biomedical Engineering, School of Life Science, Shanghai University, Shanghai 200444, China; 3School of Communication and Information Engineering, Shanghai University, Shanghai 200444, China

**Keywords:** Tau PET, radiomics, Alzheimer’s Disease, Mild Cognitive Impairment

## Abstract

Alzheimer’s Disease (AD) and Mild Cognitive Impairment (MCI) are closely associated with Tau proteins accumulation. In this study, we aimed to implement radiomics analysis to discover high-order features from pathological biomarker and improve the classification accuracy based on Tau PET images. Two cross-racial independent cohorts from the ADNI database (121 AD patients, 197 MCI patients and 211 normal control (NC) subjects) and Huashan hospital (44 AD patients, 33 MCI patients and 36 NC subjects) were enrolled. The radiomics features of Tau PET imaging of AD related brain regions were computed for classification using a support vector machine (SVM) model. The radiomics model was trained and validated in the ADNI cohort and tested in the Huashan hospital cohort. The standard uptake value ratio (SUVR) and clinical scores model were also performed to compared with radiomics analysis. Additionally, we explored the possibility of using Tau PET radiomics features as a good biomarker to make binary identification of Tau-negative MCI versus Tau-positive MCI or apolipoprotein E (ApoE) ε4 carrier versus ApoE ε4 non-carrier. We found that the radiomics model demonstrated best classification performance in differentiating AD/MCI patients and NC in comparison to SUVR and clinical scores models, with an accuracy of 84.8 ± 4.5%, 73.1 ± 3.6% in the ANDI cohort. Moreover, the radiomics model also demonstrated greater performance in diagnosing AD than other methods in the Huashan hospital cohort, with an accuracy of 81.9 ± 6.1%. In addition, the radiomics model also showed the satisfactory classification performance in the MCI-tau subgroup experiment (72.3 ± 3.5%, 71.9 ± 3.6% and 63.7 ± 5.9%) and in the MCI-ApoE subgroup experiment (73.5 ± 4.3%, 70.1 ± 3.9% and 62.5 ± 5.4%). In conclusion, our study showed that based on Tau PET radiomics analysis has the potential to guide and facilitate clinical diagnosis, further providing evidence for identifying the risk factors in MCI patients.

## 1. Introduction

Alzheimer’s Disease (AD) is a common neurodegenerative disease marked by chronic primary progressive memory decline and cognitive impairment, which is one of the most serious diseases threatening the elderly [1]. At present, the early identification and accurate diagnosis for prodromal AD are crucial for clinical decision-making and future development of treatments. Mild Cognitive Impairment (MCI), as a prodromal stage of AD, remains the most common underlying AD pathology or mixed pathology [2]. In line with the latest A-T-N framework, pathologic Tau is closely associated with neurodegeneration and necessary for AD-related downstream events [3,4,5]. Quantifiable tau loads and its corresponding increase may be a relevant target engagement marker for clinical disease-modifying interventions in anti-Tau agents.

Positron emission tomography (PET) offers the opportunity for non-invasively detecting regional distribution of Tau pathology at early stages of neurodegenerative disorders. First-generation Tau PET ligands have been developed as a highly credible biomarker of 3R/4R Tau deposits [6]. For instance, ^18^F-flortaucipir (known as ^18^F-AV-1451) PET pattern in AD/MCI specifically targets the clinically affected brain regions (e.g., medial temporal and lateral temporoparietal regions) and shows a strong regional association with domain-specific neuropsychological tests [7]. New Tau PET ligands (e.g., ^18^F-MK-6240, ^18^F-PI-2620 and ^18^F-Florzolotau (also known as ^18^F-APN-1607 and ^18^F-PM-PBB3) overcome the off-target binding of the first-generation products and provide fresh insight on the time course of Tau accumulation related to other biomarkers and clinical manifestation [8,9]. The application of qualitative and quantitative measure of Tau PET imaging, on the other hand, is in its early stages. The existing PET biomarker and corresponding “defined cutoffs” may not always reflect the presence or absence of pathology. One Tau-negative study estimate that 27.5% of MCI or dementia due to AD in those >75 years of age might be Tau-PET negative [10]. At this time, it is unknown how much pathologic Tau can be present in the brain below the in vivo Tau PET detectable threshold. As the most popular qualitative and quantitative analysis for PET imaging, visual reading and standard uptake value ratio (SUVR) may necessitate the sacrifice for complete information in relation to underlying regional Tau protein deposition. We anticipate that minimal neurofibrillary changes that are detectable by neuropathology examination can also be identified by Tau PET. Moreover, some studies have confirmed that brain Tau PET signal changes with age in cognitively unimpaired individuals and AD patients [11,12,13]. Tau pathology accumulates early in aging and relentlessly progresses in the course of AD. These limitations bring challenges to the clinical utilization of Tau PET imaging.

Radiomics analysis can be applied to explore previously unrecognized signs and patterns of disease evolution and progression by transforming image data into high-throughput features that are difficult to detect by the visual system or intensity-based metrics [14]. Until now, it has been applied to a variety of neuropsychiatric diseases including AD/MCI. Previous studies including MRI, ^18^F- fluoro-2-deoxyglucose (^18^F-FDG) PET and Amyloid β-protein (Aβ) PET have shown that radiomics features and classification models have potential as biomarkers for the diagnosis of AD and MCI [15,16,17,18]. These provide important imaging information for the heterogeneity distribution of microstructure, metabolism and pathological Aβ in AD or MCI. However, there is no similar research to deeply explore Tau neuropathological profile. It is also debatable whether radiomics analysis can be employed in Tau-negative PET images. The apolipoprotein E (ApoE) ε4 gene has been identified as a significant genetic risk factor for AD/MCI [19]. Previous results found associations between the gene expression and the deposition of Tau for AD [19]. The relationships between Tau PET radiomics features and genetic expression are not well understood.

Considering the important role of Tau deposits in clinical symptoms and pathological revelations [20] and the ability of radiomics in high-throughput mining of image features, we hypothesizes that based on Tau PET radiomics analysis may also be dynamic in the classification of AD and MCI patients. Furthermore, we anticipate that this method will be used as neuroimaging biomarkers to differentiate patients with risk factors. Hence, the first objective of this study is to propose and validate Tau-based radiomics features model for diagnosing AD/MCI patients by different cohorts (Alzheimer Disease Neuroimaging Initiative (ADNI)-Huashan hospital) and different Tau PET tracers (^18^F-AV1451-^18^F-Florzolotau). Additionally, we explored the possibility of using radiomics features as a good biomarker to make binary identification of Tau-negative MCI versus Tau-positive MCI or ApoE ε4 carrier versus ApoE ε4 non-carrier, which is of significant importance, but limited for clinical tests.

## 2. Materials and Methods

Figure 1 shows the overall workflow of Tau PET radiomics analysis, namely, (A) collection of images and division of subgroups, (B) image preprocessing, (C) identification regions of interest (ROIs), (D) feature extraction and selection and (E) SVM classification.

### 2.1. Subjects

All subjects were collected from two different cohorts: ADNI database and Huashan hospital, Fudan university. (1) For ADNI cohort, 121 AD patients, 197 MCI patients and 211 normal control (NC) subjects were enrolled from ADNI-1, ADNI-2, ADNI-3 and ANDI GO. Detailed subject inclusion information for ADNI cohort can be found at http://adni.loni.usc.edu (accessed on 3 May 2022). (2) For Huashan hospital cohort, 44 AD patients, 33 MCI patients and 36 NC subjects were enrolled. AD or MCI patients from Huashan hospital were clinically evaluated and judged by senior neurologists of cognitive disorders based on the current diagnostic guidelines [21,22]. NC subjects had no history for neurologic and psychiatric disorders, and no abnormal neurological examination.

For ADNI and Huashan hospital cohort, age, gender, years of education and Mini-Mental State Examination (MMSE) score were recorded. Imaging data, including ^18^F-flortaucipir (ADNI only) PET, ^18^F-florzolotau PET (Huashan hospital only) and T1-weighted structural MRI were collected. Table 1 shows the basic characteristics of all the subjects.

The ADNI cohort was approved by the institutional review board at each site and all the participants provided their written consent. The institutional review board of Huashan Hospital (HIRB) granted ethics approval for Huashan hospital cohort (No. 2018-363). All patients from Huashan hospital provided written informed consent.

### 2.2. Radiomics Model

Image Acquisition and preprocessing

Subjects in ADNI and Huashan cohort were scanned by structural T1 MRI and Tau PET. Detailed information about the ANDI acquisition protocol is described on the website (http://adni.loni.usc.edu/ accessed on 3 May 2022). Participants from Huashan hospital underwent a 3.0-T anatomical MRI (Discovery MR750; GE Medical Systems, Milwaukee, WI, USA) with FOV = 25.6 cm, matrix = 256 × 256 × 152, slice thickness = 1 mm, repetition time (TR) = 8.2 ms, echo time (TE) = 3.2 ms, flip angle= 12°. 18F-Florzolotau PET were acquired on a Siemens mCT Flow PET/CT scanner (Siemens, Erlangen, Germany) in three-dimensional (3D) mode over a 20 min acquisition time (90–110 min) and reconstructed by the ordered subset expectation maximization (OSEM) method. The detailed acquisition protocol for Huanshan hospital has been reported in our previous study [23].

All PET images preprocessing were performed in MATLAB R2018a (MathWorks, Natick, MA, USA) using the Statistical Parametric Mapping toolbox (version 12; http://www.fil.ion.ucl.ac.uk/spm/software/spm12/ accessed on 9 May 2022). Frist, PET images were co-registered with corresponding T1-weighted MRI images. Second, co-registered PET images were normalized to the Montreal Neurological Institute (MNI) space using the forward the spatial transformation matrix. Third, normalized PET images were subsequently smoothed with a Gaussian kernel with a full width at half maximum of 8 mm to blur image edges and improve the signal-to-noise ratio.

Definition of ROIs

For Tau PET, we concentrated on brain areas associated with AD-related Tau protein deposition, and defined these ROIs to obtain more detailed radiomics features. Namely, a group comparison using a two-sample t test between AD and NC from ANDI training datasets (including 85 AD patients and 148 NC subjects) were performed to define the ROIs with significant differences (FDR corrected, *p* < 0.01 and cluster size > 500). These ROIs were mapped to Automated Anatomical Labeling (AAL) for localization by xjView9.6 (http://www.alivelearn.net/xjview accessed on 23 May 2022). As MCI remains the most common underlying AD pathology or mixed pathology, we assume that these ROIs overlap MCI-related brain areas and can also be used to extract MCI radiomics features. Furthermore, the AD related regions were considered as ROIs to maintain consistency of radiomics analysis in subsequent studies.

Radiomics Feature Extraction and Selection

For each subject, radiomics features from each AD related ROIs were computed by a MATLAB toolkit for radiomics analysis (https://github.com/mvallieres/radiomics/ accessed on 6 June 2022). First, the Lloyd-Max quantization algorithm was applied to normalize the preprocessed PET images for isotropic resampling. Second, radiomics features were calculated from quantized PET images. Finally, 3 features from first-order histogram, 9 features from the Gray-Level Co-occurrence Matrix (GLCM), 13 features from the Gray-Level Run-Length Matrix (GLRLM), 13 features from the Gray-Level Size Zone Matrix (GLSZM) and 5 features from the Neighborhood Gray-Tone Difference Matrix (NGTDM) were extracted. Global features were extracted from the intensity histogram of the ROIs, whereas GLCM, GLRLM, GLSZM and NGTDM textures are matrix-based features. The detailed mathematical definition of the radiomics matrices were previously reported [18].

After feature extraction, two steps were performed for features selection: (1) Correlation analysis was first performed to reduce the dimensionality. If the correlation coefficient of two feature columns exceeded 0.1, we removed one of them randomly. (2) Second, a two-sample student’s t test between AD and NC from ANDI training datasets (including 85 AD patients and 148 NC subjects) were used to further select the features with significant differences (*p* < 0.005).

Classification

The subjects from ADNI data were randomly assigned to training and validation datasets at proportions of 0.7 and 0.3, respectively. The SVM was applied to construct the classification models of the AD-NC and MCI-NC groups based on the selected features with five-fold cross-validation 100 times in training datasets and the validation dataset was used to verify the robustness of our radiomics model. Then, the data from Huashan hospital were used as independent external test sets to validate the reliability and robustness of the corresponding models. In addition, age and sex had been treated as the covariates for SVM classification. Receiver operating characteristic (ROC) curves and the corresponding areas under the curve (AUC) were used to evaluate the diagnostic capabilities of the radiomics features.

### 2.3. Comparative Models

To verify the superiority of radiomics model, two comparative models were performed as the followed: (1) SUVR model: the SUVR value of each ROI was calculated by a reference region (cerebellum) and used as the input of the classifier. (2) Clinical scores model: MMSE scores, as the inputs, were construct the clinical prediction model. The SVM with a linear kernel function was also used as the classifier in the comparative experiment.

### 2.4. Radiomics Model in MCI Subgroups

To explore the performance of radiomics model on the identification of Tau-negative MCI vs. Tau-positive MCI or MCI ApoE ε4 carrier vs. ApoE ε4 non-carrier, MCI patients were further divided into subgroups. (1) For Tau-negative MCI vs. Tau-positive MCI(MCI-tau (+)/MCI-tau(-)), MCI Tau PET images were visually interpreted by two experienced neuroimaging specialist who were blinded to clinical information and made positive or negative decisions based global cortical binding. The final binary decision was based on the consensus of two independent assessors. (2) For ApoE ε4 carrier vs. ApoE ε4 non-carrier (MCI-ApoE(+)/MCI-ApoE(-)), ApoE gene expression was recorded only in 171 MCI patients from ANDI cohort. The ApoE status was determined by the ApoE ε4 gene expression or not. Radiomics model was treated with the same method as above.

### 2.5. Statistical Analysis

Statistical analyses were performed using SPSS software 26.0 (IBM Corporation, Armonk, NY, USA). For categorical and continuous variables, the demographic information was collected as numbers or means ± SD. The chi-squared tests for categorical variables (sex) and one-way ANOVA test between AD, MCI and NC groups was performed. Values were considered significant for *p* < 0.05.

## 3. Results

### 3.1. Demographic and Clinical Characteristics

The demographic and clinical characteristics of the ANDI cohort and Huashan hospital subjects are presented in Table 1. (1) For the ANDI cohort, there was a significant difference in age, sex, years of education and MMSE between AD and NC or MCI and NC group (*p* < 0.05) and the AD group is different from MCI group in MMSE scores (*p* < 0.05). There is no difference in age, sex or years of education between AD and MCI group. (2) For Huashan hospital cohort, a difference in age, years of education and MMSE between MCI and NC group (*p* < 0.05) and a difference in years of education and MMSE between AD and NC group (*p* < 0.05) and there was a difference in age and MMSE between AD and MCI group (*p* < 0.05). There is no age difference between AD and MCI group. No difference was found in sex among the AD, MCI and NC group.

### 3.2. The Defined ROIs and Selected Features

In final, 60 ROIs based on AAL atlas were obtained from the above method (Appendix A). The result showed that majority of the ROIs were found in the frontal, temporal and occipital lobe (Figure 2).

The total amount of features extracted from ROIs was 2580 ((3 + 40) × 60 = 2580). After the features selection, 31 features mainly from GLSZM and NGTDM were left in the frontal, temporal and occipital lobe. The details of these features provided in Appendix A.

### 3.3. Tau PET Radiomics Model for the Diagnosis AD/MCI

For the identification of AD from NC, we obtained an accuracy of 84.8 ± 4.5% with the ADNI validation dataset by radiomics model and an accuracy of 81.9 ± 6.1% with the Huashan hospital as the independent external test data. The performances of the SUVR and Clinical scores model were poorer than radiomics model with accuracies of 80.3 ± 1.4% and 70.5 ± 5.2%, respectively, in the ADNI validation dataset and 75.1 ± 3.5% and 66.4 ± 10.2%, respectively, in the Huashan hospital cohort (Table 2).

For the identification of MCI from NC, we obtained an accuracy of 73.1 ± 3.6% with the ADNI validation dataset. The performances of the SUVR and Clinical scores model were poorer than radiomics model with accuracies of 70.8 ± 2.7% and 65.1 ± 5.2%, respectively, in ADNI validation dataset. The accuracy with the Huashan hospital as the independent external test data was 63.5 ± 8.7%. The performances of Clinical scores model (accuracy: 63.1 ± 11.0%) were very similar to radiomics model in the Huashan hospital. However, the performances of the SUVR model (accuracy: 68.7 ± 5.5%) were not poorer than radiomics model (Table 3).

Compared to SUVR or Clinical scores model, the median AUC of the radiomics model reached 0.906/0.850 and achieved the best performance for diagnosis AD/MCI from NC (Figure 3).

### 3.4. Tau PET Radiomics Model for the Diagnosis MCI Subgroups

With the MCI-tau(+) vs. NC classification, we obtained an accuracy of 93.5 ± 2.7% and 72.3 ± 3.5% for the ADNI training data and validation data, respectively. With the MCI-tau(-) vs. NC classification, the accuracy in the training data and validation data was 91.7 ± 0.9% and 71.9 ± 3.6%, respectively. The performance of the MCI-tau(+) vs. MCI-tau(-) classification was also excellent with the accuracies of 83.4 ± 5.2% and 63.7 ± 5.9% in ADNI training data and validation data, respectively (Table 4.). The AUC for MCI-tau(+) vs. NC, MCI-tau(-) vs. NC and MCI-tau(+) and MCI-tau(-) were 0.918 (0.829–0.955), 0.820 (0.752–0.907) and 0.711 (0.668–0.805), respectively (Figure 4).

For the identification of MCI-ApoE(+) from NC, we obtained an accuracy of 92.7 ± 1.1% and 73.5 ± 4.3% with the ADNI training data and validation data, respectively. For the identification of MCI-ApoE(-) from NC, we obtained an accuracy of 92.5 ± 2.9% and 70.1 ± 3.9% with the ADNI training data and validation data, respectively. In addition, we obtained an accuracy of 87.1 ± 8.9% and 62.5 ± 5.4% for the classification of MCI-ApoE(+) vs. MCI-ApoE(-) (Table 5). The AUC for MCI-ApoE(+) vs. NC, MCI-ApoE(-) vs. NC and MCI-ApoE(+) and MCI-ApoE(-) were 0.910 (0.861–0.937), 0.826 (0.788–0.853) and 0.701 (0.632–0.747), respectively (Figure 4).

## 4. Discussion

So far, few studies had investigated the use of artificial intelligence on Tau PET images for the assessment of neurodegenerative diseases. In this paper, we proposed Tau PET-based radiomics analysis as a novel biomarker to apply to AD/MCI. Meanwhile, we selected two cross-racial independent cohorts with different PET scanners, two imaging tracers, to prove the stability and generalization of the method. We find that this radiomics model has the potential of improving the diagnostic accuracy for AD/MCI, even contributing to the identification of MCI with negative or positive Tau PET. Moreover, we evaluated this model could predict the ApoE4 carrier results of MCI patients, which is an important risk factor predicting progression to dementia.

Radiomics seeks to extract high-throughput quantitative information from medical images, especially those that are difficult for the human eyes to recognize or quantify [14]. Prior studies offered solid evidence that AD/MCI patients had Tau deposition in the frontal, temporal, parietal and occipital lobes [24,25]. In our study, AD-related ROIs were characterized by SPM analysis in frontal, temporal and occipital lobe, which is consistent with those reported in the above literature. Eventually, 31 radiomics features, mainly from GLSZM and NGTDM, in the temporal, parietal, occipital lobes and cingulate gyrus were left. The GLSZM-derived features assess the variability of gray-level intensity values and the distribution of large area size zones in the image [26]. The NGTDM-derived features mainly reflect the difference between a gray value and the average gray value of its neighbors [26]. These radiomics features were usually difficult to detect by manual inspection, but computer-aided technology scan effectively identified them. Significant differences on the above features showed the highest inter-patient variability within the distributions of voxel values. Additionally, it provided multidimensional evidence that Tau deposit occurred in specific brain regions.

Currently, more evidence highlighted the possibility that radiomics can be employed as imaging biomarkers for AD and MCI [27,28]. T1-weighted Magnetic Resonance Imaging (MRI) radiomics methods were first used to distinguish AD/MCI from NC. Other MRI sequences, including Voxel-Based Morphometry (VBM), Susceptibility-Weighted Imaging (SWI) and Diffusion Tensor Imaging (DTI), were used in detecting the brain structural and functional changes of AD and MCI [27]. For example, Feng et al., performed the logistic analysis with a classification accuracy of 0.9 for AD vs. NC, an accuracy of 0.81 for AD vs. MCI and an accuracy of 0.75 for MCI vs. NC [29]. For FDG PET, radiomics features provided the best performance with classification accuracy of 0.77 vary to 0.94on MCI/NC and AD/NC [18,30]. As the common Aβ neurobiological biomarkers, the high-order features of Aβ PET also achieved an accuracy of 0.87 for AD vs. NC classification [31]. Compared with above studies, our Tau PET radiomics model achieved similar to classification accuracy. Additionally, the classification accuracy remained slightly lower in independent external test dataset from the Huashan hospital cohort. Notably, the Tau PET tracer in Huashan hospital cohort is different from ANDI cohort. Thus, we can conclude that the high accuracy achieved was a consequence of the robustness of the radiomics classification model. According to our results of the comparative experiment, the performance of this model outperformed SUVR or Clinical scores model. For Tau PET, SUVR typically defined as the ratio of average activity in brain ROI relative to reference (usually in cerebellum). However, the reference in cerebellum has some disadvantages including small size, low signal detection sensitivity and the partial volume effect (PVE) [32]. For MMSE scores, it has shown not to be adequate in detecting MCI and clinical signs of dementia due to the ceiling or the floor effect and higher subjectivity [33]. Hence, the incomplete characteristics of the SUVR, limitation of the neuropsychological scales may lead to the comparative results [34].

Variations in the types, amounts and distribution of concomitant AD or non-AD pathologies may account for the Tau ‘positivity’ or ‘negativity’ of MCI [35]. Previous studies showed that these Tau negativity individuals were less likely to have AD-related clinical features and that the majority did not develop dementia over at least 5 years of follow-up [36]. Early in vivo diagnosis of MCI with Tau positivity, which may evolve into AD, is critical for accurate patient management. In our study, the radiomics models exhibited satisfactory performance in automated detection of MCI with Tau-negative or Tau-positive cases with mean accuracy of 72.3% or 71.9% from NC. This method could be helpful to identify and eventually treat patients as early as possible in the disease process. It also could be applied to overcome obvious shortcomings of traditional assessment, such as manual operations of image intensity and inter-reader variability of visual interpretation.

The APOE ε4 genotype expression is related to higher risk of AD/MCI [37]. The associations between the genetic phenotypes and AD-associated Tau deposition had been proven and light the genetic basis for Tau deposition [38]. Considering the toxicity, identification of APOE ε4 carriers and blocking its action may delay or stop the development of AD [39]. As expected, our study showed that radiomics features was also affected by the ApoE ε4 genotype. This radiomics model showed the high accuracy for the identification of APOE ε4 carrier or non-carrier from NC. It is meaningful that Tau radiomics features had been confirmed to have genetic significance and were helpful for identifying MCI with risk genetic factor.

For this study, we draw attention to some limitations. First, the diagnosis of AD/MCI was not confirmed by the autopsy. AD is a significant heterogeneous disease with various forms clinical presentation, which is now referred to as the Alzheimer spectrum [40]. We strictly adopted the standardized clinical diagnostic criteria to classify patients into AD and MCI. Second, we divided MCI subjects into MCI-tau(+) and MCI-tau(-) group by visual interpretation. Considering subjectivity of the naked eye, a reliable strategy for tau PET analysis is desired to be developed in the future. Third, we did not use the scale for related exclusion study. Whether the bias of the scale has an impact on the results needs further discussion. Fourth, we only employed single independent external cohorts with relatively small the number of subjects. A larger cohort and a multicenter study is required for stronger verification in future research. Finally, the study is its retrospective nature. Ongoing longitudinal observational studies in the model will be explored to validate these results.

## 5. Conclusions

In conclusion, we explored radiomics model for the classification of AD/MCI based on Tau deposition. Our results demonstrated that this model could acquire high-level evidence for clinical practice and accurately and stably identify AD/MCI from NC. In addition, we also find that these radiomics features can identify the risk factors in MCI patients, i.e., deposition of Tau and APOE ε4 gene expression. These findings show that Tau PET radionics can serve as new neuroimaging biomarker for clinical aided classification, further providing evidence that advanced machine learning methods may contribute to clarify the neuropathological mechanism for AD from a new perspective.

## Figures and Tables

**Figure 1 brainsci-13-00367-f001:**
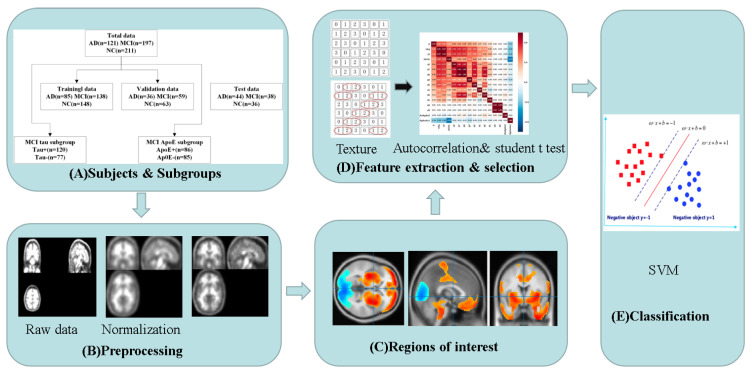
The main workflow for Tau PET radiomics analysis comprised five sections: subjects, subgroups, preprocessing, regions of interest and classification. SVM: support vector machine.

**Figure 2 brainsci-13-00367-f002:**
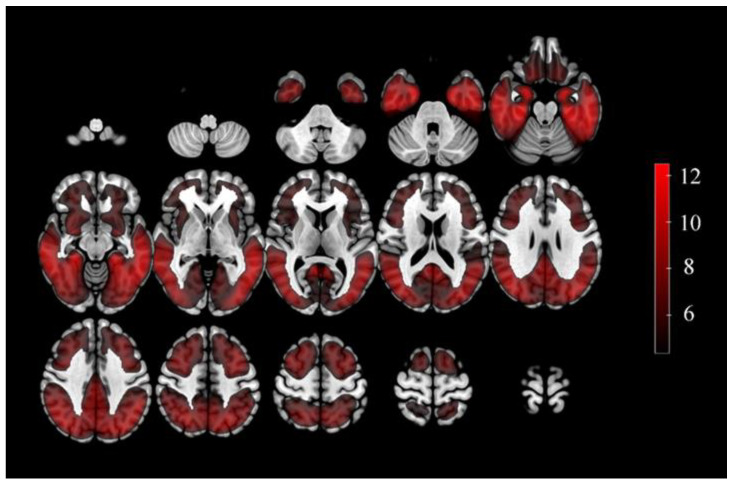
The ROIs related brain regions defined by a two-sample t test between AD and NC from ANDI training datasets. Color bars represent t value. ROIs: Regions of interest.

**Figure 3 brainsci-13-00367-f003:**
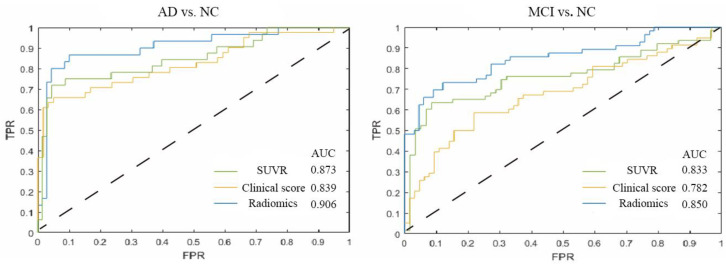
Receiver operating characteristic (ROC) curves in classification of AD vs. NC [AUC: SUVR 0.873 (0.847–0.913), Clinical score 0.839 (0.796–0.962), Radiomics 0.906 (0.850–0.933)] and MCI vs. NC [AUC: SUVR 0.833 (0.797–0.859), Clinical score 0.782 (0.729–0.851), Radiomics 0.850 (0.802–0.911)]. Data are given as median (interquartile range). TPR: True Positive Rate; FPR: False Positive Rate; AUC: Areas under the curve.

**Figure 4 brainsci-13-00367-f004:**
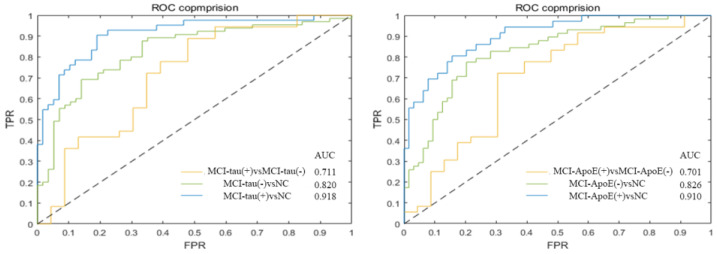
Receiver operating characteristic (ROC) curves in classification of MCI-tau subgroups and MCI-ApoE subgroups. [MCI-tau(+)-MCI-tau(-) AUC: 0.711 (0.668–0.805), MCI-tau(-)-NC AUC: 0.820 (0.752–0.907), MCI-tau(+)-NC AUC: 0.918 (0.829–0.955)]; [MCI-ApoE(+)-MCI-ApoE(-) AUC: 0.701 (0.632–0.747), MCI-ApoE(-)-NC AUC: 0.826 (0.788–0.853) and MCI-ApoE(+)-NC AUC: 0.910 (0.861–0.937)]. Data are given as median (interquartile range). TPR: True Positive Rate; FPR: False Positive Rate; AUC: Areas under the curve.

**Table 1 brainsci-13-00367-t001:** Demographic, clinical characteristics for ANDI cohort and Huashan hospital subjects.

	Age (Years)	Sex (Male/Female)	Education (Years)	MMSE
ANDI cohort				
**AD (*n* = 121)**	**72.1 ± 7.5 ***	**55/66 ***	**15.5 ± 2.6 ***	**24.0 ± 3.3 ***
**MCI (*n* = 197)**	**71.1 ± 7.4 ^†^**	**108/89 ^†^**	**16.4 ± 2.5 ^†^**	**27.9 ± 1.9 ^†‡^**
**NC (*n* = 211)**	**71.2 ± 6.4**	**79/132**	**16.7 ± 2.3**	**29.1 ± 1.2**
Huashan hospital				
**AD (*n* = 44)**	**58.2 ± 9.6**	**17/27**	**9.8 ± 4.2 ***	**16.6 ± 6.9 ***
**MCI (*n* = 33)**	**69.4 ± 8.4 ^†‡^**	**10/23**	**10.4 ± 3.2 ^†^**	**25.6 ± 1.8 ^†‡^**
**NC (*n* = 36)**	**58.5 ± 8.2**	**18/20**	**10.1 ± 2.1**	**27.2 ± 2.5**

Data are given as numbers or mean ± standard deviation (SD) values. * *p* < 0.05 AD vs. NC. † *p* < 0.05 MCI vs. NC. ‡ *p* < 0.05 AD vs. MCI. MMSE: Mini-Mental State Examination.

**Table 2 brainsci-13-00367-t002:** The classification results for AD vs. NC subjects.

Model	Accuracy (%)	Sensibility (%)	Specificity (%)
Radiomics			
**Validation**	**84.8 ± 4.5**	**76.1 ± 5.1**	**88.7 ± 2.9**
**Test**	**81.9 ± 6.1**	**83.8 ± 4.9**	**78.6 ± 7.3**
SUVR			
**Validation**	**80.3 ± 1.4**	**61.5 ± 3.5**	**87.0 ± 5.0**
**Test**	**75.1 ± 3.5**	**60.8 ± 7.8**	**79.1 ± 1.5**
Clinical scores			
**Validation**	**70.5 ± 5.2**	**58.2 ± 13.9**	**79.9 ± 12.0**
**Test**	**66.4 ± 10.2**	**53.3 ± 6.5**	**70.2 ± 11.7**

**Table 3 brainsci-13-00367-t003:** The classification results for MCI vs. NC subjects.

Model	Accuracy (%)	Sensibility (%)	Specificity (%)
Radiomics			
**Validation**	**73.1 ± 3.6**	**71.3 ± 6.1**	**75.0 ± 5.5**
**Test**	**63.5 ± 8.7**	**65.7 ± 8.8**	**60.6 ± 5.8**
SUVR			
**Validation**	**70.8 ± 2.7**	**58.5 ± 14.8**	**88.4 ± 9.8**
**Test**	**68.7 ± 5.5**	**53.8 ± 14.4**	**86.7 ± 8.7**
Clinical scores			
**Validation**	**65.1 ± 5.2**	**42.5 ± 13.9**	**87.5 ± 12.0**
**Test**	**63.1 ± 11.0**	**49.8 ± 9.6**	**80.5 ± 21.5**

**Table 4 brainsci-13-00367-t004:** The classification results for MCI-tau subgroups.

	Accuracy (%)	Sensibility (%)	Specificity (%)
MCI-tau(+) vs. NC			
**Train**	**93.5 ± 2.7**	**92.0 ± 2.2**	**94.1 ± 3.6**
**Validation**	**72.3 ± 3.5**	**70.4 ± 5.9**	**74.0 ± 5.8**
MCI-tau(-) vs. NC			
**Train**	**91.7 ± 0.9**	**91.0 ± 1.2**	**92.0 ± 3.4**
**Validation**	**71.9 ± 3.6**	**70.1 ± 6.0**	**73.5 ± 5.1**
MCI-tau(+) vs. MCI-tau(+)			
**Train**	**83.4 ± 5.2**	**88.5 ± 7.3**	**80.1 ± 4.5**
**Validation**	**63.7 ± 5.9**	**69.4 ± 6.6**	**53.2 ± 8.0**

**Table 5 brainsci-13-00367-t005:** The classification results for MCI-ApoE subgroups.

	Accuracy (%)	Sensibility (%)	Specificity (%)
MCI-ApoE(+) vs. NC			
**Train**	**92.7 ± 1.1**	**92.7 ± 2.1**	**93.8 ± 1.8**
**Validation**	**73.5 ± 4.3**	**68.0 ± 3.8**	**76.6 ± 4.6**
MCI-ApoE(-) vs. NC			
**Train**	**92.5 ± 2.9**	**91.0 ± 3.3**	**92.9 ± 2.0**
**Validation**	**70.1 ± 3.9**	**68.0 ± 3.0**	**72.8 ± 5.1**
MCI-ApoE(+) vs. MCI-ApoE(-)			
**Train**	**87.1 ± 8.9**	**90.3 ± 10.5**	**83.6 ± 5.7**
**Validation**	**62.5 ± 5.4**	**71.6 ± 7.2**	**51.6 ± 11.0**

## Data Availability

The datasets generated during and/or analyzed during the current study are available from the corresponding author on reasonable request.

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
