# Peer review of "Based on Tau PET Radiomics Analysis for the Classification of Alzheimer’s Disease and Mild Cognitive Impairment"

_brainsci, 2023, doi:10.3390/brainsci13020367_

Round 1

Reviewer 1 Report

This manuscript entitled “Based on Tau PET Radiomics Analysis for the Classification of Alzheimer's Disease and Mild Cognitive Impairment” provides radiomics analysis to discover high-order features from pathological biomarker and improve the classification accuracy based on

Tau PET images.  The topic that the authors have worked to analyze is a very important one deserving of a comprehensive work, and recognition. Apart from minute typographical errors (spelling and capitalization) that needs rectification, I have few suggestions to authors.

1.      Figure 1 is a flow diagram, so it should look like a flow chart.

2.      Formatting needs special attention, many times spaces between paragraph and within lines in not same.

3.      Make figure legends more informative.

4.     a general oversimplification of the discussion is done. 

Reviewer 2 Report

The authors in this article determined “Based on Tau PET Radiomics Analysis for the Classification of Alzheimer's Disease and Mild Cognitive Impairment”. In the present research article, the authors have aimed to implement radiomics analysis to discover high-order features from pathological biomarker and improve the classification accuracy based on Tau PET images. Methods: Two crossracial independent cohorts from the ADNI database (121 AD patients, 197 MCI patients and 211 normal control (NC) subjects) and Huashan hospital (44 AD patients, 33 MCI patients and 36 NC subjects) were enrolled. The radiomics features of Tau PET imaging of AD related brain regions were computed for classification using a support vector machine (SVM) model. The radiomics model was trained and validated in the ADNI cohort and tested in the Huashan hospital cohort. Standard uptake value ratio (SUVR) and clinical scores model were also performed to compared with radiomics analysis. Additionally, we explored the possibility of using Tau PET radiomics features as a good biomarker to make binary identification of Tau-negative MCI versus Tau-positive MCI or ApoE ε4 carrier versus ApoE ε4 non-carrier. Results: The radiomics model demonstrated best classification performance in differentiating AD/MCI patients and NC in comparison to SUVR and clinical scores models, with an accuracy of 84.8% ± 4.5%, 73.1% ± 3.6% in the ANDI cohort. Moreover, the radiomics model also demonstrated greater performance in diagnosing AD than other methods in the Huashan hospital cohort, with an accuracy of 81.9% ± 6.1%. In addition, the radiomics model also showed the satisfactory classification performance in the MCI tau subgroup experiment (72.3% ±3.5%,71.9% ± 3.6% and 63.7% ±5.9%) and in the MCI-ApoE sub-group experiment (73.5% ±4.3%,70.1% ± 3.9% and 62.5% ±5.4%).Conclusions: Our study showed that based on Tau PET radiomics analysis has the potential to guide and facilitate clinical diagnosis, further providing evidence for identifying the risk factors in MCI patients.

I can see very few articles in the present topic, which adds advantage for this study to be novel even though there are some flaws in the hypothesis and results of the article. I would like to recommend some major concerns to the authors to fulfil the hypothesis, as there are scarce data for sample size and results in the article.

v  In introduction, the authors have elaborately written the introduction, but details about your hypothesis and objectives need to be elaborate and clearly explained. https://doi.org/10.2174/1874467214666210906125318 Kindly check.

v  In materials and method several experiments are not clearly explained, please refer https://doi.org/10.3389/fmolb.2022.1050768, https://doi.org/10.1186/s12929-022-00871-6, https://doi.org/10.3389/fmolb.2022.1030534, please correct.

v  The authors need to explain how they did the exclusion of the related studies using the scale, I could not see the scale criteria’s, they can give the scale in a schematic illustration format. The given format is not clear.

v  The scoring scale of the  structural T1 MRI and Tau PET of the study and the patient classification need to be discussed.  

v  The authors need to show the final schematic conclusion diagram revealing a summarised concluding evidence and mechanism.

v  The authors need to check the abbreviations, in first time usage they need to use the full word and abbreviation. A separate section for abbreviation would be better.

v  The authors need to concentrate in the requested changes as per review comments and check the manuscript for some grammatical errors and mistakes.

The available research information seems to be insufficient, and the authors need to address the above comments. Taking together to all this issue I recommend major revision to the manuscript in present form.

Reviewer 3 Report

This paper proposes a radiomics analysis to discover high-order features from pathological biomarkers and improve the classification accuracy based on Tau PET images. My comments are as follows:

1-    The abstract should be a single paragraph and should follow the style of structured abstracts but without headings. 

2 - Some new references should be added to improve the literature review—for example, https://doi.org/10.1007/s00247-022-05510-8; https://doi.org/10.1155/2020/1357853.

Overall, this can be a useful work, and some interesting aspects are presented. I recommend that the authors should revise their paper according to the above-addressed points.

Round 2

Reviewer 2 Report

Most of my previous comments has been answered and the manuscript has been significantly improved. The research article is well written and have discussed the points pertaining their novelty and creates scientific interest for the readers. The available research information seems to be sufficient and advised for publication. In current form I recommend the manuscript can be published in Brain Sciences.